# Locating Core Modules through the Association between Software Source Structure and Execution

**Sang Moo Huh** [1] and **Woo Je Kim** [2,*]

1   Graduate School of Public Policy and Information Technology, Seoul National University of Science and Technology, Gongneung-ro 232, Nowon-gu, Seoul 01811, Korea; norhuh@naver.com
2   Department of Industrial Engineering, Seoul National University of Science and Technology, Gongneung-ro 232, Nowon-gu, Seoul 01811, Korea
*   Correspondence: wjkim@seoultech.ac.kr; Tel.: +82-2-970-6449

**Abstract:** To improve software quality, the source code that composes software has to be improved, and improving the important code that largely affects the software quality should be a cost-effective method. Static analysis defines important codes as those that occupy important positions in the source network, while dynamic analysis defines important codes as those with high execution frequency. However, neither method analyzes the association between network structure and execution frequency, and both have their disadvantages. Thus, this study analyzed the association between source network structure and execution frequency to solve their disadvantages. The source function of Notepad++ was analyzed, and the function ranking was derived using the association between network structure and execution frequency. For verification, the Spearman correlation between the newly derived function ranking and the function ranking of the network and execution frequency obtained with the conventional method was measured. By measuring the Spearman correlation, the newly derived function ranking had strong correlations with execution frequency and included the network structure's characteristics. Moreover, similar to the Pareto principle, the analysis showed that 20% of Notepad++'s functions could be categorized as important functions, largely affecting the software's quality.

**Keywords:** data envelopment analysis (DEA); social network analysis (SNA); software profiling; execution frequency; Pareto principle; Notepad++; software quality



## 1. Introduction

Many companies use software to provide useful services to their customers. If defects occur in the software, companies may suffer immense damage. Therefore, removing software defects has become an essential activity for companies, and because software continues to become more extensive and complex, companies want to remove defects cost effectively. A cost-effective method for removing defects involves managing the important sources that largely affect the software service. The Pareto principle of software testing suggests that 80% of defects occur in 20% of the source code [1]. Subsequently, managing important source codes has been presented as a cost-effective method [2]. If developers can determine the important source codes that largely affect software quality, they can improve software quality cost effectively [3]. A conventional method of deriving important source codes is static analysis, which derives code that occupies an important position in the source code's network structure [2]. The method is bound to an approximate solution, leading to overapproximation and numerous false positives [3,4]. The second method is dynamic analysis, which derives frequently executed modules by running software similar to the actual core module because it uses software execution [2]. However, underapproximation is disadvantageous when important modules occupying an important network position cannot be derived and return as false negatives [3,4]. Neither method analyzes the association between network structure and execution frequency, which leads

to problems from which important functions are inaccurately derived. This study identifies important source codes by linking network structure and execution frequency to solve the problem. Section 1 considers the necessity of this study and previous literature, and Section 2 explains the applied methodology and research materials. Section 3 derives the results of the study, Section 4 evaluates the differences from previous literature and the paper, and Section 5 reviews the significance of the paper and future research.

Feature location technology utilizes static, dynamic, and hybrid analysis techniques to track codes that implement feature sets [3–19]. The static analysis of feature location identifies the relevant code using the source's call structure without executing the software. Chen and Rajlich [5] proposed using absolute system dependence graphs (ASDGs) to identify static function locations, where the user looks for the relevant code based on the system dependencies. Robillard [6] introduced a more automated static analysis approach that examines the topology of system dependencies. Meanwhile, Saul et al. [7] applied a hypertext-induced topic selection (HITS) algorithm to subsets of static call graphs. For dynamic analysis, methods such as a software profiling technique, software reconnaissance [4,8–10], and execution slice [11] track the execution unit. Two sets of scenarios (those with and without functions) are executed, and the relevant code is identified through differences between the set of executed units [4,8,10–14]. Hybrid techniques such as single trace and information retrieval (SITIR) [15] and probabilistic ranking of methods based on execution scenarios and information retrieval (PROMESIR) [16] combine text analysis and dynamic analysis. Eisenbarth, Koschke, and Simon [12] applied a formal, conceptual analysis to the execution trace and combined the results with an approach similar to ASDG [17]. On the other hand, Hill, Pollock, and Vijay-Shanker [18], as well as Zhao et al. [19], conducted studies that incorporate text and static analysis information, which are useful when finding relevant codes for specific functions. However, these were inconsistent with this study's aim of locating important functions in the overall software.

Some studies analyzed the software's network structure using social network analysis (SNA), which calculates the weight of nodes and links using various indicators of complex networks [20,21]. SNA could detect the dependency of defects in dynamic-link libraries, leading to predicting defects with 60% accuracy [6,22,23]. Li and Yi [24] derived the class of the unified modeling language's weight using the PageRank social network technique, and Sarkani [25] prioritized components by graphically analyzing component connections for testing in component-based systems. Hu [26] applied social network techniques to Eclipse, Netbeans, and Gnome's Java packages, Java Archive files, and Ubuntu packages to derive claims for the data set and is strongly associated with defect prediction. Using SNA, Nguyen, Adams, and Hassan [27] predicted class and package defects. He et al. [28] measured the defect tendencies and the severity of modified defects at the class level for Apache Tomcat, arguing that the importance of classes can frequently predict defects. Furthermore, Rahman [1] utilized SNA technology to derive JFreeChart and JHotDraw's network class prestige and the Eclipse tool's program to measure their execution frequencies, and also used four SNA indicators to analyze the Spearman correlation between each class's prestige and execution frequencies. Huh and Kim [29] derived eight types of core functions using eight SNA indicators for the CUBRID open database source, deriving the weighted priority of the integrated functions by integrating these into data envelopment analysis (DEA) technique.

The software profiling technique allows the measurement of the execution frequency of software functions [30,31]. Renieres and Reiss [30] helped localize errors by using the execution frequency of multiple lines of code in a program spectrum form. Harrold et al. [31] studied empirical methods of comparing program behaviors by analyzing the program spectrum and the related branch-count spectra, which are the conditional branches of the program's execution frequency.

This study aims to find important modules by analyzing the association between the software's network structure and the execution frequency. In a feature location field, techniques combining static and dynamic analysis were investigated, and similar studies

were examined. However, research combining static and dynamic analysis information that satisfies the purpose of this study was insufficient.

DEA, a nonparametric method of measuring efficiency, can analyze the relationship between the software network structure derived by SNA and the execution frequency derived by profiling technique, which uses mathematical programming rather than regression [32–35]. This method can handle multiple inputs and outputs for each operational unit without assuming functional relationships. As it measures the relative efficiency of decision-making units (DMUs) using multiple input and output elements, DEA can calculate the functions' integrated weights, and those with high weighted values are more likely to be actual core functions [32–35]. This study derives the software function ranking through a combination of SNA, software profiling, and DEA techniques to verify whether the shortcomings of existing studies have improved and derives the core functions that affect software quality. This study discusses the different hypotheses for checking the correlation between the static and dynamic analyses' results.

## 2. Materials and Methods

### 2.1. Social Network Analysis (SNA)

A social network consists of connected nodes and links, and these connections are the fan-in and fan-out of nodes forming a network in a software system. The structure and characteristics of the software network are analyzed with a graph to model the relationships between modules. If nodes occupying important network positions are located, the entire network's effectiveness increases through intensive node management [20,21]. As the source modules of software call each other and form a network, social network indicators can locate core modules occupying important network positions.

Degree centrality is the first SNA indicator, which indicates how many connections a node has. Simply put, the connectedness of a node is expressed as the number of connections made to the node; the more connections a node has, the higher the degree of centrality. Connectedness considers several components: in-degree, out-degree, and total degree [20,21]. In-degree is when a node is called (fan-in), while out-degree is when a node calls another node (fan-out). The connectedness of a node ($V_i$) is expressed as:

$$Cd_{(V_i)} = d_i \tag{1}$$

$$Cd_{(V_i)} = d_i^{(in)} \tag{2}$$

$$Cd_{(V_i)} = d_i^{(out)} \tag{3}$$

$$Cd_{(V_i)} = d_i^{(in)} + d_i^{(out)} \tag{4}$$

$d_i$ = the number of connections made to the node $v_i$.

Eigencentrality, the second SNA indicator, measures the importance of a node relative to its neighboring nodes' importance. In special cases, eigencentrality values may be zero. To avoid obtaining a zero value, the PageRank algorithm—a variant of eigencentrality used by Google—estimates the importance of a node by counting the number of links and measuring the linked nodes' quality. Accordingly, the importance of a node propagates to all nodes connected to it [24]. PageRank is expressed as:

$$Cp_{(V_i)} = \frac{(1-d)}{n} + d \sum_{i=1}^{n} \frac{PR(T_i)}{C(T_i)} \tag{5}$$

Cp (V) = PageRank score of node;
$V_i T_i$ = node that points to the $V_i$ node;
PR ($T_i$) = PageRank score of node $T_i$;
C ($T_i$) = the number of outgoing links from node $T_i$;
d = damping factor (0.5 was applied in this study).

The third SNA indicator, HITS, assigns an authority and hub weight to a node. A node has a higher authority weight if nodes with high hub weights are linked to it. Similarly, a higher hub weight occurs if one node is linked to multiple nodes with higher authority weights. HITS can analyze referral relationships among nodes to identify which occupy important positions in the network structure. Equations (6) and (7) calculate the authority and hub weights of the third indicator [35].

$$\text{Hub}[n] = \sum_{\forall n', n - \text{po int} - n'} \text{Auth}[n'] \tag{6}$$

$$\text{Auth}[n] = \sum_{\forall n', n - \text{po int} - n'} \text{Hub}[n'] \tag{7}$$

Because measuring a node's influence in an entire network by solely deriving its importance using direct connections is insufficient, the derived important nodes will utilize direct and indirect connections [2,21,36]. The closeness centrality measures the sum of the entire network's path distances to derive the most central node, which was selected as the fourth indicator for analysis as this was most likely to be frequently executed. The betweenness indicator, which is the study's fifth indicator, derives functions that passes through to reach another function, which is likely to be frequently called.

### 2.2. Software Profiling

Each code's execution frequency is an internal software attribute that can measure the actual value at runtime, and software profiling can measure the execution frequency at different granularity levels [37]. Execution frequency measures for statements, branches, and functions [30,38], and the information provided by this technique is widely used to evaluate test case quality, optimize program performance, or detect memory leaks. In this study, software profiling was used to measure the execution frequencies of software source modules.

### 2.3. Data Envelopment Analysis (DEA)

Despite its unconventional application, previous studies used DEA to evaluate software quality, performance, productivity, and efficiency [32–35,39–46]. In this study, DEA was used to analyze the association between SNA and software profiling results. Modules occupying important positions in the network, and the execution frequencies of modules identified through software profiling, were derived using DEA by measuring the efficiency of the DMUs. When there are several DMUs (n), inputs (i) and outputs (j) utilizing the Charnes, Cooper, and Rhodes (CCR) model with Equations (8)–(10) can obtain the DMUs' relative efficiencies [32].

$$\text{Efficiency} = \frac{\text{The weighted sum of outputs}}{\text{The weighted sum of inputs}} \tag{8}$$

$$\text{Max } E_k = \frac{\sum_{r=1}^{s} u_r y_{kr}}{\sum_{i=1}^{m} v_i x_{ki}} \tag{9}$$

$$\text{Subject to } E_k = \frac{\sum_{r=1}^{s} y_{jr} u_{kr}}{\sum_{i=1}^{m} x_{ji} v_{ki}} \leq 1, \; j = 1, 2, \ldots, n \tag{10}$$

$$\mu_{kr} \geq \varepsilon, r = 1, 2 \ldots, s$$
$$v_{ki} \geq \varepsilon, \; i = 1, 2 \ldots, m$$

$E_k$ = DMU$_k$ efficiency;
$u_{kr}$ = weights given to DMU$_k$ output r (r = 1, 2, . . . , s);
$y_{kr}$ = DMU$_k$ output r (r = 1, 2, . . . , s);
$v_{ki}$ = weights given to DMU$_k$ input i (i = 1, 2, . . . , m);
$x_{ki}$ = DMU$_k$ input i (i = 1, 2, . . . , m);

$\varepsilon$: a very small value, $0 \le E_k \le 1$;

n: the number of DMUs;

m: the number of inputs;

s: the number of outputs.

DEA models are classified into either the CCR or the Banker, Charnes, and Cooper (BCC) models depending on the returns to scale (RTS). The CCR model estimates variable returns, wherein the relationship between inputs and outputs varies depending on the scale [32]. In 1984, Banker, Charnes, and Cooper [33] developed the BCC model, wherein the relationship between inputs and outputs can be estimated at certain rates, regardless of the scale [34,35].

### 2.4. Demonstration of Node Ranking Combining Network Structure and Execution Frequency by DEA

DEA explains how nodes are ranked by combining the network structure and execution frequency information expressed in the network structure (Figure 1). SNA indicators examined the network structure, and the node's weight was presented differently for each indicator (Table 1). The software profiling technique presumably measured the execution frequency as a numerical value indicated above the node in Figure 1, as shown in Table 1.

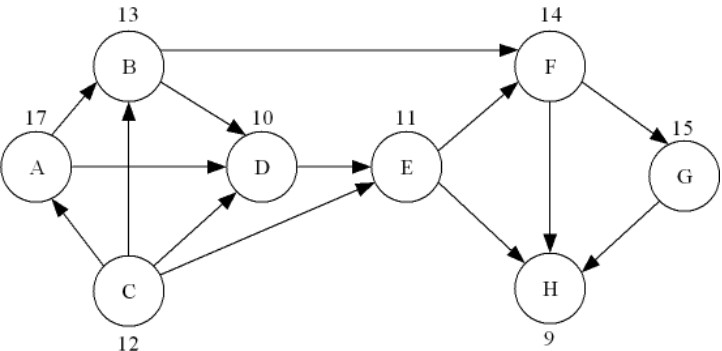

**Figure 1.** Sample network and execution frequency of nodes.

**Table 1.** The analytical value of the SNA and execution frequencies of nodes in Figure 1.

| Node | Static Analysis (Social Network Analysis (SNA)) | | | | | | | | Dynamic Analysis |
|------|-----------|------------|--------------|-----------|--------|----------------|-----------------|-------------|---------------------|
| | In-degree | Out-degree | Page Rank | Authority | Hub | In-closeness | Out-closeness | Betweenness | Execution Frequency |
| A | 0.1429 | 0.1257 | 0.0703 | 0.3193 | 0.4867 | 0.1429 | 0.4286 | 0 | 17 |
| B | 0.2857 | 0.2857 | 0.0879 | 0.5195 | 0.3501 | 0.2857 | 0.4464 | 0.0833 | 13 |
| C | 0 | 0.5714 | 0.0625 | 0 | 0.7762 | 0 | 0.6364 | 0 | 11 |
| D | 0.4286 | 0.1429 | 0.1099 | 0.6635 | 0.1581 | 0.4286 | 0.2857 | 0.0595 | 10 |
| E | 0.2857 | 0.2857 | 0.1252 | 0.3843 | 0.1056 | 0.3810 | 0.3214 | 0.1310 | 11 |
| F | 0.2857 | 0.2857 | 0.1158 | 0.1875 | 0.0343 | 0.4464 | 0.2857 | 0.1548 | 14 |
| G | 0.1429 | 0.1429 | 0.0914 | 0.0141 | 0.0285 | 0.3673 | 0.1429 | 0 | 15 |
| H | 0.4286 | 0 | 0.1685 | 0.0693 | 0 | 0.5833 | 0 | 0 | 9 |

The DEA Equations (8)–(10) were applied to associate the network structure and execution frequency. For the network location's importance of node to satisfy the execution frequency, the node efficiency was calculated as the maximum. The network structure and execution frequency were associated using this method, and the derived node rankings are shown in Table 2.

**Table 2.** The efficiency and rank of nodes by data envelopment analysis (DEA).

| Nodes | Efficiency | Rank |
|-------|-----------|------|
| A | 73.7% | 6 |
| D | 70.% | 7 |
| C | 540.0% | 2 |
| D | 67.8% | 8 |
| E | 84.9% | 4 |
| F | 1000.0% | 1 |
| G | 84.6% | 5 |
| H | 96.4% | 3 |

*2.5. Research Procedure*

Software is developed in various languages, such as C, C++, Java, Pascal, and Python, and there are various types of software modules, such as methods, functions, procedures, classes, and objects. In this study, C++ functions that are often used to create solutions were studied. Figure 2 shows the overall research procedure. Phase (1) derives the function ranking by associating network structure and execution frequency. Meanwhile, phase (2) verifies whether the newly derived function ranking has both network and execution frequency characteristics through the hypothesis.

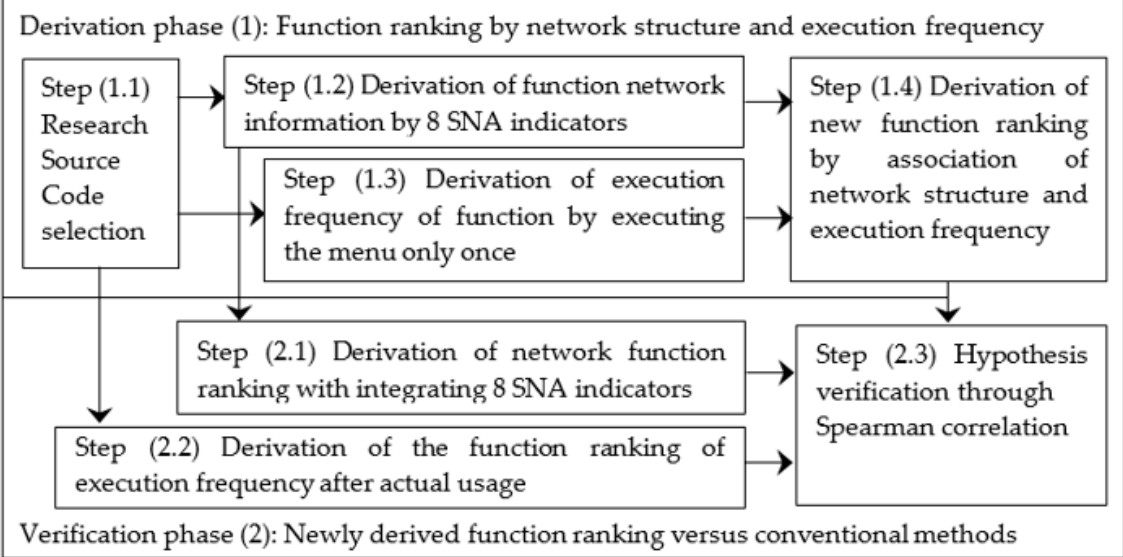

**Figure 2.** Overall research procedure.

*2.6. Derivation Phase (1): Function Ranking by Network Structure and Execution Frequency*

This phase is consistent with the study's purpose, and new source function rankings are derived through the association between software network structure and execution frequency.

2.6.1. Step (1.1): Research Source Code Selection

Research software requires an adequately sized source code that can perform all of its menu functions. In this study, operating systems and databases too large to run all menu functions and small-scale software were deemed unsuitable. However, Notepad++ version 10.0.0.136, a well-known open-source software that can execute all menu functions, was selected for the study. Analysis of the selected Notepad++ source revealed about 120 C++ source files, 1754 functions, and 243 paths. However, some functions were not called, while other functions were called solely from external systems. If these functions are included and analyzed, incorrect results would be analyzed [2]. Therefore, these functions were excluded, and the remaining 378 functions formed a network (Figure 3).

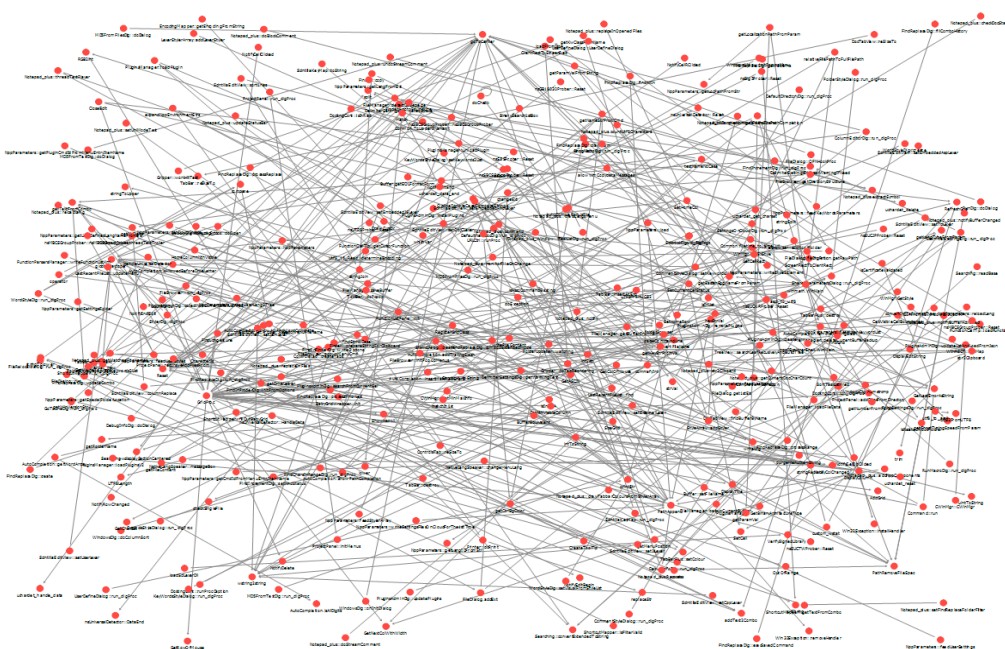

**Figure 3.** C++ source function network of Notepad++.

2.6.2. Step (1.2) (Static Analysis): Derivation of Function Network Information by 8 SNA Indicators

A total of 8 SNA indicators were used to analyze a network of 378 functions, and the network weights of the functions were derived for each SNA indicator. Here, functions with higher weights are considered important functions, and the source functions' network weights differ for each social network indicator (Table 3).

**Table 3.** Part of the network weight of 378 functions analyzed by 8 SNA indicators.

| Function | 8 Social Network Analysis (SNA) Indicators | | | | | | | |
|---|---|---|---|---|---|---|---|---|
| | In-degree | Out-degree | Page-Rank | Authority | Hub | In-closeness | Out-closeness | Betweenness |
| DisplayColumn | 2653 | 13,263 | 1342 | 164,373 | 130,571 | 2653 | 13,642 | 4 |
| DisplayEditString | 7958 | 2653 | 1895 | 196,486 | 30,884 | 8488 | 3537 | 35 |
| DisplayTitle | 2653 | 0 | 1342 | 164,373 | 0 | 2653 | 0 | 0 |
| doCheck | 2653 | 0 | 3968 | 0 | 0 | 2653 | 0 | 0 |
| DockingCont::isInRect | 0 | 2653 | 1323 | 0 | 0 | 0 | 2653 | 0 |
| DockingCont::run_dlgProc | 0 | 2653 | 1323 | 0 | 0 | 0 | 2653 | 0 |
| DockingCont::runProcCaption | 0 | 2653 | 1323 | 0 | 0 | 0 | 2653 | 0 |
| DocTabView:: findBufferByName | 0 | 2653 | 1323 | 0 | 0 | 0 | 3537 | 0 |
| DocTabView::reSizeTo | 0 | 2653 | 1323 | 0 | 0 | 0 | 2653 | 0 |
| doException | 2653 | 2653 | 1383 | 0 | 0 | 2653 | 2653 | 7 |
| DrawCursor | 2653 | 2653 | 1342 | 164,373 | 30,884 | 2653 | 3537 | 0 |

2.6.3. Step (1.3) (Dynamic Analysis): Derivation of Execution Frequency of Function by Executing the Menu Only Once

The Notepad++ functions' source codes were modified to measure the execution frequency of the C++ function. When measuring the execution frequency, if the same menu is executed multiple times, the source functions related to the menu may be deemed as more important. In this case, measurements may not accurately analyze the association between the network structure and the execution frequency. Therefore, every menu in Notepad++ was executed only once before measuring the source functions' execution frequencies (Table 4).

**Table 4.** Part of the execution frequency of 378 functions measured by running every menu once.

| Functions | Execution Frequency |
|---|---|
| Notepad_plus::process | 188,207 |
| nsBig5Prober::Reset | 20,865 |
| StyleArray::addStyler | 15,276 |
| Notepad_plus::command | 12,113 |
| getKwClassFromName | 9176 |
| Shortcut::run_dlgProc | 8962 |
| getNameStrFromCmd | 8769 |
| FindReplaceDlg::run_dlgProc | 6340 |
| LexerStylerArray::addLexerStyler | 6100 |
| FindGrid | 5266 |
| GridProc | 5238 |
| commafyInt | 3626 |

2.6.4. Step (1.4): Derivation of New Function Ranking by the Association of Network Structure and Execution Frequency

Function importance was calculated using Equations (8)–(10) of the DEA, which satisfied its execution frequency by maximizing the function's efficiency with 8 network weights' information. The network information and execution frequency can be associated using this method, and the new function ranking was derived (Table 5).

**Table 5.** The part of 378 functions ranking calculated by the DEA technique.

| New Functions Ranking | Functions |
|---|---|
| 1 | Notepad_plus::process |
| 2 | GetWinVersionStr |
| 3 | CalcVisibleCellBoundaries |
| 4 | Notepad_plus::str2Cliboard |
| 5 | Notepad_plus::replaceInFiles |
| 6 | GetCharset |
| 7 | CreateToolTip |
| 8 | nsSBCSGroupProber::Reset |
| 9 | checkSingleFile |
| 10 | FunctionCallTip::getCursorFunction |
| 11 | ScintillaKeyMap::run_dlgProc |
| 12 | CommentStyleDialog::run_dlgProc |
| 13 | ClientRectToScreenRect |
| 14 | Searching::convertExtendedToString |
| 15 | NppParameters::feedUserLang |

*2.7. Verification Phase (2): Newly Derived Function Ranking versus Conventional Methods*

It is ideal that the newly derived function ranking is compared with the actual function ranking, but the actual function ranking is unknown. Therefore, the study determined if the derived result was correlated with the execution frequency and the network's function rankings. Following this step, the top 20% of all functions selected as core functions were analyzed.

2.7.1. Step (2.1): Derivation of Network Function Ranking with Integrating the Results of 8 SNA Indicators

The network weights of the functions analyzed by the SNA indicators are shown in Table 3, but the representative network weight was unknown. In previous studies, several indicators were integrated into one using the DEA technique [29]. By applying this method, the functions' network weights were integrated into 1, and the ranking distribution of its weight was shown in Figure 4a. By sorting them in the highest weight order, the integrated network ranking of the function was derived.

### 2.7.2. Step (2.2): Derivation of Function Ranking of Execution Frequency after Actual Usage

Notepad++, which can measure the execution frequency, was distributed to several users, and several execution frequencies of the source function were collected after usage. Then, the function ranking was derived using the collected execution frequencies. After analyzing the correlation between the function ranking of the execution frequency and the newly derived function rankings in this study, most were similar. Because the correlations are similar, the three-frequency function ranking types (highest, medium, and lowest correlation) are described, as shown in Figure 4b–d.

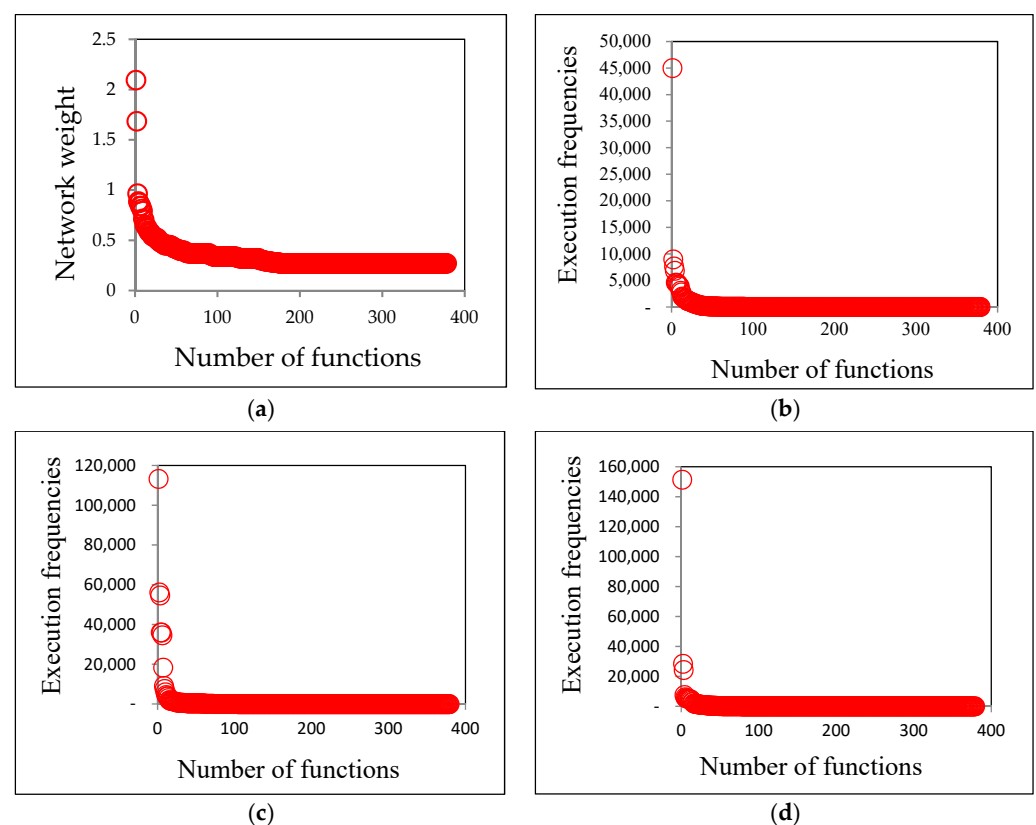

**Figure 4.** The function ranking distribution of integrated network weight and three execution frequencies; (**a**) Ranking distribution of integrated network weight; (**b**) Ranking distribution of the execution frequency with the highest correlation (H); (**c**) Ranking distribution of the execution frequency with the medium correlation (M); (**d**) Ranking distribution of the execution frequency with the lowest correlation (L).

### 2.7.3. Step (2.3): Hypothesis Verification through Spearman Correlation

Because the integrated network and execution frequency function rankings were derived, the study's hypothesis was tested using the Spearman correlation, which compared the correlation of function rankings. The following hypotheses were established to determine whether the conventional technique's disadvantages were resolved, and 20% of the core functions could be adopted similar to the Pareto principle.

**Hypothesis 0.** *When the conventional technique was used, there is a correlation between the network function ranking using static analysis and the function ranking of execution frequency using dynamic analysis.*

**Hypothesis 1.** *The new function ranking derived in this study has no correlation with the network function ranking.*

**Hypothesis 2.** *The new function ranking derived in this study has no correlation with the execution frequency function ranking.*

**Hypothesis 3.** *The top 20% of all functions derived from this study can be adopted as core functions.*

The correlation between the static analysis and dynamic analysis results was analyzed before verifying this study's results. Based on Hypothesis 0, if the network function ranking and the execution frequency function ranking are correlated, the other hypotheses may be meaningless. Hypothesis 1 verifies the correlation between the new function rankings derived in this study and the network function rankings, and Hypothesis 2 verifies the correlation with the execution frequency's function ranking. Moreover, Hypothesis 3 analyzes whether the top 20% of all functions can be selected as core functions.

## 3. Results

### 3.1. Hypothesis 0 Test

The Spearman correlation between the network and execution frequency function rankings was analyzed, as shown in Table 6. The Spearman correlation was different for each of the eight SNA indicators, and there was a weak correlation between the integrated network function ranking and the execution frequency of medium (M) and lowest data (L) in Figure 4. However, the execution frequency of highest data (H) was analyzed with a $p$-value > 0.05, which was not related to the integrated network function ranking. Therefore, Hypothesis 0 was rejected because there was no correlation between the network and execution frequency function rankings.

**Table 6.** Spearman correlation between the function ranking of network and execution frequency.

| Spearman Correlation Comparison | | | Figure 4b Highest Correlation Data | Figure 4c Medium Correlation Data | Figure 4d Lowest Correlation Data |
|---|---|---|---|---|---|
| Table 3 | Degree Centrality | In-degree | 8.9% | 10.6% | 13.5% |
| | | Out-degree | −0.5% | −1.7% | −1.9% |
| | HITS Centrality | Authority | 6.5% | 16.0% | 16.3% |
| | | Hub | 5.0% | 10.3% | 10.% |
| | Closeness Centrality | In-closeness | 9.7% | 11.2% | 13.9% |
| | | Out-closeness | 2.0% | 0.6% | 0.2% |
| | Page Rank centrality | | 9.5% | 10% | 12.4% |
| | Node betweenness | | 7.4% | 7.2% | 9.7% |
| Figure 4a | Integrated 8 SNA indicators | | 0.94% | 12.1% | 13% |

### 3.2. Hypothesis 1 Test

The Spearman correlation between the newly derived function ranking and the network function ranking was analyzed, as shown in Table 7, and the correlation of eight SNA indicators was analyzed differently. The integrated eight SNAs, including all of the indicators' characteristics, can be selected as a representative indicator [29]. The $p$-value of the integrated SNA index was greater than 0.05 and had a 25.58% Spearman correlation. Therefore, Hypothesis 1 was rejected.

**Table 7.** The correlation between the derived function ranking and the network function ranking.

| Spearman Correlation Comparison | | | Spearman Correlation | $p$-Value | Hypothesis 1 |
|---|---|---|---|---|---|
| Table 3 | Degree Centrality | In-degree | 24.596% | $1.293 \times 10^{-6}$ | Rejected |
| | | Out-degree | −9.413% | 0.6754 | Not Rejected |
| | HITS Centrality | Authority | 16.708% | 0.001112 | Rejected |
| | | Hub | 13.074% | 0.01095 | Rejected |
| | Closeness Centrality | In-closeness | 24.549% | $1.362 \times 10^{-6}$ | Rejected |
| | | Out-closeness | −7.775% | 0.1313 | Not Rejected |
| | PageRank centrality | | 23.239% | $4.97 \times 10^{-6}$ | Rejected |
| | Node betweenness | | 15.125% | 0.0032 | Rejected |
| Figure 4a | Integrated 8 SNA indicators | | 25.58% | $4.63 \times 10^{-7}$ | Rejected |

### 3.3. Hypothesis 2 Test

The Spearman correlation between the newly derived function ranking and the execution frequency's function rankings after actual usage was analyzed, as shown in Table 8, and the *p*-values of the execution frequency were greater than 0.05. Hypothesis 2 was rejected because the Spearman correlation between the newly derived function ranking and the execution frequency's function ranking ranged from 79.6–83.12%.

**Table 8.** The correlation between the function ranking of execution frequency after actual usage.

| Ranking of Execution Frequency | Spearman Correlation | *p*-Value | Hypothesis 2 |
|---|---|---|---|
| Highest correlation data | 83.121% | $2.2 \times 10^{-16}$ | Rejected |
| Medium correlation data | 82.169% | $2.2 \times 10^{-16}$ | Rejected |
| Lowest correlation data | 79.699% | $2.2 \times 10^{-16}$ | Rejected |

### 3.4. Hypothesis 3 Test

The Spearman correlations of the newly derived function rankings relative to the function rankings of the execution frequency and the integrated network were presented in Figure 5. In the graph, the newly derived function ranking was set as the *x*-axis, the Spearman correlation percent was set as the *y*-axis, and the Spearman correlations between the execution frequency and the network were drawn as line graphs. There was no Spearman correlation under 10% of the *x*-axis, whereas, above 10%, Spearman correlations were present. When comparing the three types of execution frequencies with the newly derived function rankings, they increased the correlation by 20%, decreased after 20%, and later increased after 57%. All had similar correlations, as shown in the line graph. Correlations with the network function ranking increased up to 57% of the *x*-axis, and there were correlations of 25% at 59% or more above the *x*-axis. When analyzing the correlation at 20% of the Pareto principle's function, it has strong correlations with the function ranking of execution frequency. There was also a strong network correlation than the correlation of the entire function. Therefore, 20% of the functions can be selected as the core function, and Hypothesis 3 was chosen.

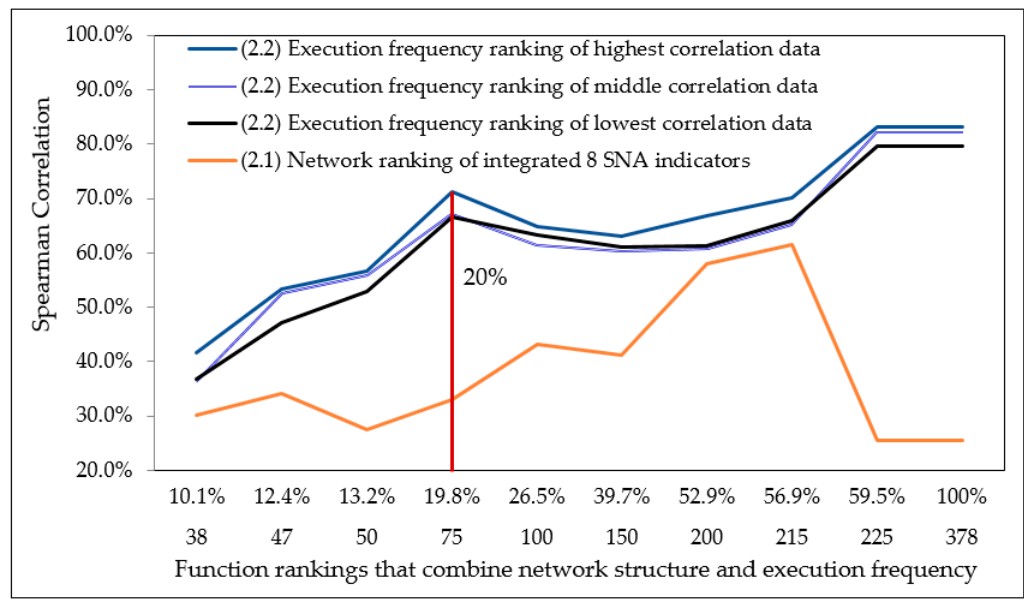

**Figure 5.** Correlation graph between the function ranking and network and execution frequency.

### 4. Discussion

This study analyzed a new method that locates source codes that largely affect software quality. Conventional methods include static analysis, which analyzes the source network

structure, and dynamic analysis, which locates frequently executed modules. One static analysis study analyzed Java classes with four SNA indicators and analyzes how much they match frequently executed classes [2]. The study concluded that there was a correlation between network structure and execution frequency in some classes. However, among the four SNA indicators, representative indicators that can be used are not presented. Huh and Kim [29] solved this problem by presenting a method that integrates the results of 8 social network indicators through the DEA technique. However, because the two studies do not analyze the association with execution, there is little association with execution [3,4]. Meanwhile, the dynamic analysis method measures frequently executed modules through execution to approximate the important modules. However, it is impossible to derive modules that are important position in the source network [3,4]. In this study, a new method of deriving function rankings by analyzing the association between the network structure and execution frequency was presented. The newly derived function rankings by this study's method had a correlation with the execution frequency's function ranking and the network function ranking. Therefore, the disadvantages of static analysis and dynamic analysis were resolved. Moreover, 20% of the functions are deemed important functions that can affect software quality, similar to the Pareto principle. These findings suggest that Notepad++'s software quality can be improved cost effectively. However, applying this research method to other software does not guarantee the same results. For this method to become applicable to other software, additional research on other software is required.

## 5. Conclusions

This study investigated a new method of deriving important source codes using the association between software network structure and execution. The significance of this study is as follows. First, source function rankings can be derived by analyzing the association between the network structure and execution frequency using the DEA technique. Second, a new association method that can derive important software modules was proposed. Third and last, a new method that derives the top 20% of functions that conform to the Pareto principle was presented. This method is expected to be applicable when studying other software, deriving core modules, and improving overall software quality cost effectively.

This study did not analyze McCabe's complexity in the source code and the relationships of functions such as inheritance, association, and dependencies. Future studies can obtain more accurate results by analyzing the functions' relationship and McCabe complexity and examining how software quality increases after improving the newly derived core functions.

**Author Contributions:** Conceptualization, S.M.H. and W.J.K.; methodology, S.M.H. and W.J.K.; validation, W.J.K.; investigation, S.M.H.; resources, S.M.H. and W.J.K.; writing—original draft preparation, S.M.H.; writing—review and editing, S.M.H. and W.J.K.; project administration, S.M.H. and W.J.K. All authors have read and agreed to the published version of the manuscript.

**Funding:** This research received no external funding.

**Institutional Review Board Statement:** Not applicable.

**Informed Consent Statement:** Not applicable.

**Data Availability Statement:** Not applicable.

**Conflicts of Interest:** The authors declare no conflict of interest.

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
