# Peer review of "Locating Core Modules through the Association between Software Source Structure and Execution"

_applsci, doi:10.3390/app11041685_

Round 1

Reviewer 1 Report

In this paper, the authors present a new association method between source network structure and execution frequency was presented to solve their disadvantages and the new C++ source functions ranking was derived by applying it to Notepad++ source.

The problem is interesting as well as the work, and it tackles a question dealt in the literature using an innovative approach. It has practical applications.
Paper is well structured.
The state of the art is sufficient, and the topic is well explained.
The description of goals is clear as well as the methodology applied.
References are appropriate.

Author Response

Thank you very much for your careful review.

Reviewer 2 Report

The work presented is appropriate to be published in this journal.

Only minor details:

  • The structure of the paper must be placed at the end of the introduction section.
  • There are some English mistakes.

Author Response

(The authors gave the same response as above.)

Reviewer 3 Report

Paper Summary.
This paper proposed a new software engineering methodology that associates reference network structure found in source code, and code execution frequency analyzed in executing apps. They have applied this method into the well-known and publicly available OSS Notepad++ and successfully found that the core functions mainly affect software quality.

Strengths.
The model definition is exact, and it contributes to the replicability of the proposed method. An application scenario looks practical.

Author Response

Thank you very much for your careful review.

Many awkward English expressions were corrected throughout the whole paper.

Reviewer 4 Report

The authors present a proposal for a new association method between source network structure and execution frequency

The article is well written. The topic is within the scope of the journal. It is easy to read.

Some improvements:
a) The new journal template should be used.
b) Regarding the structure of the article, a discussion section is necessary in which the results obtained with other similar works are compared and the progress of the study carried out can be assessed.

Author Response

(The authors gave the same response as above.)

Round 2

Reviewer 4 Report

The paper can be accepted in current form